# MMBERT: A MODERN MULTILINGUAL ENCODER WITH ANNEALED LANGUAGE LEARNING

## ABSTRACT

Encoder-only languages models are frequently used for a variety of standard machine learning tasks, including classification and retrieval. However, there has been a lack of recent research for encoder models, especially with respect to multilingual models. We introduce MMBERT, an encoder-only language model pretrained on 3T tokens of multilingual text in over 1800 languages. To build MMBERT we introduce several novel elements, including an inverse mask ratio schedule and an inverse temperature sampling ratio. We add over 1700 low-resource languages to the data mix only during the decay phase, showing that it boosts performance dramatically and maximizes the gains from the relatively small amount of training data. Despite only including these low-resource languages in the short decay phase we achieve similar classification performance to models like OpenAI's o3 and Google's Gemini 2.5 Pro. Overall, we show that MMBERT significantly outperforms the previous generation of models on classification and retrieval tasks – on both high and low-resource languages.

## 1 INTRODUCTION

Encoder-only language models (LMs) were developed during the the early years of scaling language model pre-training, including models such as BERT (Devlin et al., 2019) and RoBERTa (Liu et al., 2019). These models were followed by multilingual variants such as mBERT (Devlin et al., 2019) and XLM-RoBERTa (also known as XLM-R) (Conneau et al., 2019). However, these models cannot generate text, and thus have fallen out of popularity in favor of larger decoder-only language models (Brown et al., 2020; Achiam et al., 2023; Team et al., 2023; Dubey et al., 2024).

Despite this, encoder-only models still are frequently used for natural language understanding (NLU) tasks, including classification, clustering, and retrieval. For many years, these older models were still the best encoder-only models available. However, there has been a recent revival of encoder-only model pretraining (Nussbaum et al., 2024; Portes et al., 2023; Warner et al., 2024; Boizard et al., 2025), bringing modern pre-training techniques for decoder-only language models to encoder-only models. There have also been new analyses showing that encoder-only models are significantly better for classification/retrieval than decoder-only models for a given size, even beating decoders an order-of-magnitude larger (Weller et al., 2025; Gisserot-Boukhlef et al., 2025).

In this encoder-only model revival there has been a conspicuous lack of large-scale multilinguality. Although it is over six years old, XLM-R is still SOTA – especially surprising when you consider the fast-paced nature of the LM field. Thus, we aim to provide a more recent improved version.

We do this by pre-training our new model suite, MMBERT, on 3T tokens of multilingual text using an architecture inspired from ModernBERT (Warner et al., 2024). We also propose novel contributions to the pre-training recipe: (1) an inverse mask schedule learning rate (high $\Rightarrow$ low), (2) an annealing language schedule (more biased $\Rightarrow$ more uniform), and (3) increasing the number of languages at each training phase ($60\rightarrow110\rightarrow1833$), allowing for maximal impact of the smaller amount of data.

MMBERT improves over XLM-R across the board, and even beats models like OpenAI's o3 (OpenAI, 2025) and Google's Gemini 2.5 Pro (Comanici et al., 2025) on low-resource languages. We show that including the low-resource languages in the decay phase enables rapid learning, boosting performance

---

Models, data, and code are available at <removed for anonymity>

| Model | Layers | Hidden Size | Intermediate Size | Heads | LR | WD | Warmup (B) | BS Warmup (B) |
|---|---|---|---|---|---|---|---|---|
| MMBERT small | 22 | 384 | 1152 | 6 | 8e-4 | 8e-5 | 4 | 100 |
| MMBERT base | 22 | 768 | 1152 | 12 | 8e-4 | 8e-5 | 3 | 60 |

**Shared Parameters:**
Vocabulary Size: 256,000 | Training Batch Size: 4.7M tokens | Max Sequence Length: 1024 → 8192
RoPE Base: 10k → 160k | Sliding Window: 128 | Global Attention Every N Layers: 3
Masking Rate Schedule: 30% → 15% → 5% | Tokenizer: Gemma 2 | Architecture: ModernBERT

Table 1: Configuration for MMBERT models (small is 140M params, base is 307M). MMBERT small initialized from Base and required LR/WD reduction to 4e-4/4e-5 after 1.2T tokens.

on these languages roughly 2x despite only using 100B tokens. Overall, MMBERT is the first model to show significant improvements over XLM-R for massively multilingual data while also introducing new techniques for multilingual LM pre-training that apply to both encoders and decoders.

## 2 RELATED WORK

**Encoder-only Models** Encoder-only models were the predominant language model in the early LM days, with ELMo (Peters et al., 2018), BERT (Devlin et al., 2019), and RoBERTa (Liu et al., 2019) being early examples of scaling language models up to the trillion token data range. Encoder-only models are still the predominant models used for classification and retrieval tasks when inference speed is important. However, decoder-only models have been scaled to the trillion parameter scale whereas encoder models typically remain less than 1 billion parameters.

The revival of encoder-only LM development was spurred by works such as MosiacBERT showing a BERT-equivalent model could be trained in under 24 hours (Portes et al., 2023; Nussbaum et al., 2024). More recently, ModernBERT (Warner et al., 2024) further scaled these recipes and showed that you could greatly improve performance. Since then there have been several more: EuroBERT focusing on 15 languages (Boizard et al., 2025), NeoBERT (Le Breton et al., 2025) on English, and Ettin showing paired open-data encoder and decoder recipes (Weller et al., 2025). However, none of these more recent models have scaled to more than 15 languages.

In the massively multilingual setting, there are very few available models: mBERT with 104 languages (Devlin et al., 2019), XLM-R with 100 languages (Conneau et al., 2019), and mGTE with 74 languages (Zhang et al., 2024). Works such as multilingual DistilBERT (Sanh et al., 2019) and multilingual MiniLM (Wang et al., 2020) have distilled from mBERT and XLM-R respectively to derive smaller variants. The more recent mGTE showed slightly improved performance over XLM-R while allowing for longer contexts, while mBERT is generally not used due to the improvements of XLM-R. Despite XLM-R's release date, it has aged very well: its design was well ahead of its era through its use of 6T training tokens, more than any other encoder-only model has ever been trained on, even to this day (including our MMBERT). However, data quality has significantly increased since 2019, allowing us to achieve higher scores with only half of the tokens (Penedo et al., 2024).

**Multilingual LMs and Datasets** Multilingual models and datasets have a long history of development for machine translation systems, ranging from bilingual models and datasets to massively multilingual systems (NLLB Team et al., 2022; Arivazhagan et al., 2019, among others). Recently, decoder-only LMs e.g. (Gemma et al., 2024; Dubey et al., 2024) have embraced multilinguality in order to increase accessibility (Comanici et al., 2025; Achiam et al., 2023). However, many of the details of these models are not available, including pre-training data. This has created an access gap between open massively multilingual systems and closed systems. From models that do release information, we know that improvements are often due to improved data quality or techniques like parallel data (Martins et al., 2025; Li et al., 2024b; Dubey et al., 2024). For massively multilingual datasets, previous work often derived from CommonCrawl (Wenzek et al., 2019). Our work also uses datasets derived from web-scale corpora, but with recent advances in filtering and langauge identification (Penedo et al., 2025; Kargaran et al., 2023) and other datasets from specialized sources.

| Category | Dataset | Pre-training | | Mid-training | | Decay Phase | |
|---|---|---|---|---|---|---|---|
| | | Tokens (B) | % | Tokens (B) | % | Tokens (B) | % |
| Code | Code (ProLong) | – | – | – | – | 2.8 | 2.7 |
| Code | Starcoder | 100.6 | 5.1 | 17.2 | 2.9 | 0.5 | 0.5 |
| Crawl | DCLM | 600.0 | 30.2 | 10.0 | 1.7 | – | – |
| Crawl | DCLM (Dolmino) | – | – | 40.0 | 6.7 | 2.0 | 2.0 |
| Crawl | FineWeb2 | 1196.6 | 60.2 | 506.7 | 84.3 | 78.5 | 76.0 |
| Instruction | Tulu Flan | 15.3 | 0.8 | 3.1 | 0.5 | 1.0 | 1.0 |
| Math | Dolmino Math | 11.2 | 0.6 | 4.3 | 0.7 | 0.5 | 0.5 |
| Reference | Books | 4.3 | 0.2 | 3.9 | 0.7 | 2.2 | 2.1 |
| Reference | Textbooks (ProLong) | – | – | – | – | 3.1 | 3.0 |
| Reference | Wikipedia (MegaWika) | 4.7 | 0.2 | 1.2 | 0.2 | 9.5 | 9.2 |
| Scientific | Arxiv | 27.8 | 1.4 | 5.4 | 0.9 | 3.3 | 3.2 |
| Scientific | PeS2o | 8.4 | 0.4 | 3.2 | 0.5 | – | – |
| Social | StackExchange | 18.6 | 0.9 | 3.0 | 0.5 | – | – |
| Social | StackExchange (Dolmino) | 1.4 | 0.1 | 2.8 | 0.5 | – | – |
| **Total** | | 1989.0 | 100.0 | 600.8 | 100.0 | 103.3 | 100.0 |

Table 2: Training data mixture across the various training stages (pre-training, mid-training, decay). Later stages use higher quality data, including the recent Dolmino (OLMo et al., 2025) and FineWeb2-HQ (Messmer et al., 2025) datasets. Dashes indicate that no data from that source was used. We trained for 2.3T tokens for pre-training, 600B for mid-training (e.g. context-extension to 8192 sequence length), and 100B for the decay phase. We sample from the dataset and repeat (or under-sample) as needed to hit the token counts used for training. Note that the decay phase included three different mixtures and the resulting weights were merged together, this includes only one version (Decay-Cont). See Appendix D for more details. If not specified, the source is Dolma v1.7.

## 3 TRAINING DETAILS

### 3.1 ARCHITECTURE

We use an identical architecture to ModernBERT, but employ the Gemma 2 tokenizer[1] (Gemma Team et al., 2024) to handle multilingual input. We use 22 layers and an 1152 intermediate dim for both base and small versions (same as ModernBERT-base), but use a hidden dimension of 768 for base and 384 for small. For the rest of the training configurations that are shared, see Table A in the Appendix.

Our base version has the same number of non-embedding parameters as ModernBERT-base (110M) but a total of 307M parameters due to the larger vocabulary. MMBERT small has 140M total parameters, with 42M non-embedding parameters. As we show in (§4.5) these architectures are also significantly faster than any previous multilingual encoder model while being the same standard sizes.

### 3.2 TRAINING DATA

We follow the Ettin recipe (Weller et al., 2025) as the only open-data ModernBERT equivalent. Crucially though, we change the source of web crawl to account for more multilingual data. Previous work such as XLM-R and mT5 (Xue et al., 2020) used a very low percentage of Engish content (5.7% for mT5). However, the highest quality data (filtered DCLM (Li et al., 2024a; OLMo et al., 2025)) only exists in English. Thus we choose to use a significantly higher percentage of English comparative to previous work (from 10% to 34% depending on the stage, see Table 10). Despite this relatively high amount of English, we see strong multilingual performance while also showing significantly improved English performance over previous massively multilingual encoders.

---

[1]During training of MMBERT the Gemma 3 tokenizer was released. We would encourage future work to pre-train using this tokenizer after modifying the pre-tokenizer to include prefix spaces, which we did not use. This likely would help for NER and POS tasks.

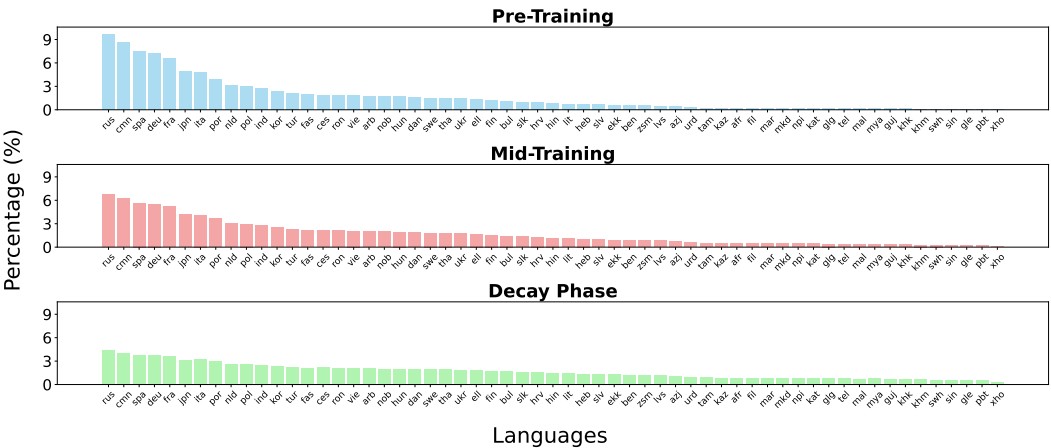

Figure 1: Our inverse temperature sampling ratio throughout training from Fineweb2 data, from $\tau$ of 0.7 to 0.5 to 0.3. Note that other sources are excluded from this chart and are not temperature sampled. For full language percent details, see Appendix D. Our training starts out more high-resource biased but becomes increasingly uniform. We also include another 50 languages in mid-training and 1723 more for the decay phase not visualized. The percent of English data used – not included in Fineweb2 – are 34.5%, 12.1%, and 10.2% for their phases respectively.

Nonetheless, a significant portion of our training data was non-English: for this we used existing data from FineWeb2 (Penedo et al., 2025) and an existing filtered version of 20 languages from FineWeb2 called FineWeb2-HQ (Messmer et al., 2025). We also use and filter MegaWika v2 (Barham et al., 2023; 2025) for Multilingual Wikipedia, which covers 60 languages (those from our first stage).

We include several other curated corpora in English. From Dolma (Soldaini et al., 2024) we use StarCoder, Stackexchange, Arxiv, and PeS2o. From Dolmino (OLMo et al., 2025) we use math, filtered Stackexchange (which is mainly code), Tulu Flan instruction data, and books (OLMo et al., 2025). From ProLong (Gao et al., 2024) we use their code repositories and their textbooks. All our data is publicly available on HuggingFace.[2] Thus, our data mix is higher quality than previous work (through the use of filtered DCLM and FineWeb2), more diverse in content (code, instructions, web data, papers), and includes a greater variety of languages and scripts.

**Cascading Annealed Language Learning (ALL)**  Unlike previous work which uses a fixed set of languages and a set temperature to sample multilingual data, we use a novel approach that changes the temperature during training and iteratively adds new languages (i.e. annealing over languages).

We observe that lower-resource languages have smaller amounts of pre-training data and relatively lower quality (since there is less to filter). Thus, we want to learn from these languages in the most impactful way (e.g. avoiding more than 5x epochs of them) while also keeping our data quality high. To do this, we start with higher resource languages and slowly add languages throughout training. At each change point, we also re-sample the ratio, taking the distribution from more high-resource biased to more uniform. This allows us to avoid doing many epochs on lower-resource data and allows for quicker learning of new languages as they already have a strong base in the existing language set (e.g. starting with Icelandic and then quickly learning Faroese).

We start with a set of 60 languages (as well as code) that cover a broad range of language families and scripts. We then increase this to 110 languages, covering more "mid-resource" languages (greater than 200 million tokens of data). Finally, we include all languages/scripts included in FineWeb2 (1833 languages, 1895 language/script variants). Our temperature for sampling languages goes from 0.7 to 0.5 to 0.3 (see Figure 1). Including all the languages at the very end allows us to take advantage of the decay phase learning to rapidly increase performance. We see this validated in Section 4.4.

---

[2]We did not create any new datasets for training, however, we did gather them and create the final data mix. We thank the open-source efforts cited above for making this possible.

| | Model Name | Single Sentence | | Paraphrase and Similarity | | | Natural Language Inference | | | |
|---|---|---|---|---|---|---|---|---|---|---|
| | | CoLA | SST-2 | MRPC | STS-B | QQP | MNLI | QNLI | RTE | Avg |
| Small | mDistilBERT | 34.7 | 89.4 | 90.5 | 87.5 | 86.6 | 79.0 | 87.5 | 73.3 | 77.5 |
| | Multilingual MiniLM | 25.4 | 91.6 | 91.3 | 88.6 | 87.7 | 82.1 | 89.7 | 75.8 | 78.3 |
| | MMBERT Small | **61.8** | **93.1** | **91.6** | **90.3** | **88.6** | **85.8** | **91.9** | **81.9** | **84.7** |
| Base | EuroBERT 210m | 36.8 | 90.6 | 92.4 | 89.8 | 88.3 | 85.3 | 91.3 | 78.3 | 81.2 |
| | XLM-R Base | 54.2 | 93.1 | 92.2 | 89.4 | 88.5 | 85.0 | 90.6 | 78.7 | 83.3 |
| | mGTE Base | 54.7 | 93.3 | **92.7** | 89.9 | 89.0 | 85.3 | 91.1 | 82.3 | 84.0 |
| | MMBERT Base | 61.9 | 94.0 | 91.9 | 91.0 | **89.4** | 87.7 | 93.3 | 85.6 | 86.3 |
| | ModernBERT Base | **65.3** | **95.3** | 90.9 | **91.5** | 89.4 | 88.8 | 93.7 | 87.7 | 87.4 |

Table 3: GLUE (English) benchmark results. We see that MMBERT outperforms all other models for their size, with the base version coming close to even ModernBERT's performance.

| | Model Name | Cross-lingual Understanding | | | Question Answering | | | Structured Prediction | | |
|---|---|---|---|---|---|---|---|---|---|---|
| | | XNLI | PAWS-X | XCOPA | XQuAD | MLQA | TyDiQA | WikiANN | UDPOS | Avg |
| Small | mDistilBERT | 60.8 | 80.2 | 52.7 | 49.4 | 43.5 | 44.2 | 54.5 | 67.1 | 56.5 |
| | Multilingual MiniLM | 71.2 | 84.6 | 59.2 | 68.2 | 56.8 | 63.0 | **59.2** | **74.2** | 67.1 |
| | MMBERT small | **73.6** | **86.7** | **61.8** | **73.0** | **62.5** | **66.7** | 54.3 | 70.6 | **68.6** |
| Base | XLM-R base | 74.6 | 85.9 | 61.2 | 73.4 | 62.1 | 70.5 | **61.4** | 74.3 | 70.4 |
| | mGTE Base | 73.9 | 86.4 | 63.6 | 75.7 | 64.3 | 69.9 | 60.7 | **74.3** | 71.1 |
| | MMBERT Base | **77.1** | **87.7** | **67.5** | **77.6** | **66.0** | **74.5** | 58.2 | 74.0 | **72.8** |

Table 4: XTREME benchmark results. Note that we exclude EuroBERT as it excludes many of the languages tested in this benchmark. For a fair comparison with EuroBERT see Section 4.2 where we compare on EuroBERT-included languages specifically.

## 3.3 TRAINING RECIPE

We use the same three phase approach as ModernBERT and Ettin but employ a novel inverse masking rate learning schedule. Rather than simply using a lower masking ratio at the end of training as shown by Boizard et al. (2025); Weller et al. (2025), we progressively lower the mask rate at each stage.[3]

For MMBERT small, we start the training process by initializing the weights from the completed base version by using strided weight sampling (Sanh et al., 2019).

**Base Pre-training** This stage encompasses the warmup and stable phase of the trapezoidal learning rate, training for 2.3T tokens. We use both learning rate and batch size warmup. The data in this stage does not include the filtered FineWeb2 data or the higher quality DCLM. We use just 60 languages (plus code languages) in this stage of training and a 30% masking rate[4] to start.

**Context Extension / Mid-Training** Here we increase the quality of the data by switching to the filtered higher-quality versions of the datasets (Table 2). We also change the RoPE values (Su et al., 2024) to handle up to 8192 tokens (i.e. theta of 160k) for global and local layers. We further increase the number of languages to 110 languages (plus code). We train for 600B tokens and continue the stable phase, lowering the mask rate to 15%.

**Decay Phase** Finally, we use an inverse square root learning rate schedule to decay for 100B tokens to 0.02 of the peak LR. Unlike ModernBERT and Ettin, we choose to decay with a 5% mask rate and use three different datasets to produce three different variants: English-focused (Decay-Eng), 110 languages (same as the mid-training phase, Decay-Cont), and a 1833 language variant (all FineWeb2 languages, Decay-All). For specific details into each mixture, see Appendix D.

**Model Merging** We then use model merging to combine the best qualities from each decay mixture. For the base version, we select the best checkpoint from each mixture and use TIES-merging (Yadav et al., 2023) to mitigate parameter interference. Merging across mixtures was ineffective for small,

---

[3] Due to the scale of pre-training, we were not able to ablate this except in the decay phase. However, we found that the smaller the masking rate the better for the decay phase of training.

[4] MMBERT small lowered the mask rate to 20% / LR to 4e-4 after 1.2T tokens when it had stopped learning.

| | Model Name | Pair Class. | Class. | STS | Retrieve | Cluster | Rerank | Summ. | Avg |
|---|---|---|---|---|---|---|---|---|---|
| Small | Multilingual MiniLM | 77.3 | 59.3 | 71.9 | 35.0 | 36.9 | 42.4 | 19.3 | 48.9 |
| | mDistilBERT | 75.9 | 59.0 | 72.4 | 38.4 | 38.3 | 42.7 | 19.4 | 49.4 |
| | MMBERT small | **78.9** | **62.1** | **74.1** | **40.7** | **40.8** | **44.1** | **23.6** | **52.1** |
| Base | EuroBERT 210m | 79.5 | 62.3 | 73.6 | 43.0 | 40.3 | 43.3 | 21.5 | 51.9 |
| | XLM-R base | 79.3 | 63.9 | 73.4 | 39.8 | 39.3 | 44.2 | 24.2 | 52.0 |
| | mGTE base | 79.9 | 64.3 | 74.4 | 43.7 | 40.2 | 43.7 | 23.0 | 52.7 |
| | ModernBERT base | **80.5** | **65.6** | **75.5** | 44.8 | **43.3** | 44.1 | 22.6 | 53.8 |
| | MMBERT base | 80.2 | 64.8 | 74.8 | **44.9** | 41.7 | **44.9** | **26.0** | **53.9** |

Table 5: MTEB v2 English results. Average is done over categories. Best model in the section is bolded. See Appendix B for sweep details. We see that both MMBERT models significantly outperform all multilingual models and MMBERT base even ties ModernBERT.

| | Model Name | Bitext Mining | Pair Class. | Class. | STS | Retrieve | Multilabel Class. | Cluster | Rerank | Avg |
|---|---|---|---|---|---|---|---|---|---|---|
| Small | mDistilBERT | 37.2 | 76.6 | 50.3 | 62.1 | 36.7 | 14.6 | 37.5 | 62.2 | 47.1 |
| | Multilingual MiniLM | 46.6 | 76.7 | **51.0** | 64.6 | 35.6 | 14.0 | 34.9 | 64.1 | 48.4 |
| | MMBERT small | **50.5** | **77.4** | 50.4 | **64.8** | **41.9** | **15.5** | **38.7** | **66.1** | **50.7** |
| Base | XLM-R base | 56.6 | 78.2 | **54.5** | 66.0 | 43.0 | 15.5 | 38.1 | 67.6 | 52.4 |
| | mGTE base | 52.6 | 78.5 | 53.1 | 66.2 | **46.4** | 17.2 | 38.8 | 69.1 | 52.7 |
| | MMBERT base | **59.2** | **79.2** | 53.6 | **67.1** | 45.8 | **17.5** | **40.2** | **69.9** | **54.1** |

Table 6: MTEB v2 multilingual results. Average is done over categories. See Appendix B for sweep details. We see that both MMBERT models significantly outperform all other models for their size.

likely due to less parameter agreement in the smaller weight space. Therefore, we merged an exponential weighting of the Decay-All checkpoints, as that merged mixture performed best.

## 4 RESULTS

### 4.1 BENCHMARK SCORES

**Datasets** We benchmark MMBERT on existing encoder benchmarks for NLU and retrieval: GLUE (Wang et al., 2018), XTREME (Hu et al., 2020), and MTEB (Enevoldsen et al., 2025). We differ from early encoder work by dropping the retrieval section of XTREME as there exist improved retrieval benchmarks from MTEB. We also include the code retrieval benchmark CoIR (Li et al., 2024c), although code is not the primary focus of MMBERT.

**Baselines** We use a variety of baselines: the older but still ubiquitous XLM-R (Conneau et al., 2019), mGTE (Zhang et al., 2024), and EuroBERT-210m (Boizard et al., 2025) for the base size and mDistilBERT (Sanh et al., 2019) and Multilingual MiniLM (Wang et al., 2020) for the small size.[5] For English, we also show results with ModernBERT (Warner et al., 2024) as an upper bound. We perform a sweep over various hyperparameters for each model, see Appendix B for details.

**NLU Results** We start with English GLUE in Table 3. MMBERT small performs significantly better than other small variants (84.7 average vs MiniLM's 78.3) and even outperforms all other previous base-sized models, including XLM-R. MMBERT base outperforms all other multilingual models, while even approaching ModernBERT's English performance (86.3 MMBERT vs 87.4 ModernBERT), despite using a majority of non-English data.

For multilingual performance in XTREME (Table 4), MMBERT base also outperforms all other models on average (72.8 average vs XLM-R's 70.4), except for structured prediction where it lags behind on POS and ties on NER. This is likely due to the same problem that ModernBERT has, i.e. the lack of a consistent prefix whitespace token during pre-training. We would recommend future work resolve this issue. However, MMBERT base shows significant improvements in classification (e.g. 77.1 XNLI accuracy vs XLM-R's 74.6) and in question-answering (74.5 F1 on TyDiQA vs 70.5

---

[5]We do not compare with RTD models. These models have consistently been shown to be significantly worse for embeddings. We validate this for mDeBERTa (He et al., 2023) in App C, where it is more than 11 pts worse.

| Task ($\rightarrow$) | | Text-to-Code | | | Code-to-Text | Code-to-Code | | | Hybrid Code | | | |
|---|---|---|---|---|---|---|---|---|---|---|---|---|
| Model (param.) $\downarrow$ | | Apps | CosQA | Synthetic Text2sql | Code SearchNet | SN-CCR | CodeTrans -Contest | -DL | StackOver Flow QA | CodeFeedBack -ST | -MT | Avg |
| Small | mDistilBERT | 2.7 | 8.4 | 35.2 | 69.4 | 31.0 | 27.5 | 21.0 | 45.8 | 37.4 | 22.8 | 30.1 |
| | Multilingual MiniLM | 2.2 | 7.6 | 22.7 | 59.8 | 21.8 | 24.9 | 22.4 | 40.4 | 35.6 | 22.6 | 26.0 |
| | MMBERT small | **4.3** | **19.5** | **41.5** | **64.1** | **38.4** | **56.3** | **32.9** | **60.3** | **52.4** | **40.7** | **41.0** |
| Base | XLM-R base | 3.1 | 13.2 | 30.5 | 75.6 | 30.7 | 34.4 | 22.1 | 51.2 | 47.3 | 28.1 | 33.6 |
| | mGTE base | 3.8 | 13.2 | 33.8 | 74.8 | 39.1 | 45.2 | 30.6 | 58.0 | 50.3 | 39.9 | 38.9 |
| | MMBERT base | **6.1** | **24.6** | **48.1** | 63.8 | 41.7 | 54.9 | 33.0 | 62.1 | 55.5 | 32.4 | 42.2 |
| | EuroBERT 210m | 5.6 | 20.3 | 43.0 | **77.0** | **43.8** | **62.2** | **35.0** | **64.4** | **57.2** | **44.6** | **45.3** |

Table 7: Retrieval scores on the CoIR Benchmark. MMBERT models outperform all others except EuroBERT, which used the higher quality but not publicly accessible Stack v2 training data.

| **XNLI** | | | | | | | | | | | | |
|---|---|---|---|---|---|---|---|---|---|---|---|---|
| **Model** | **ar** | **de** | **en** | **es** | **fr** | **hi** | **ja** | **ru** | **tr** | **vi** | **zh** | **Avg** |
| EuroBERT 210m | 66.8 | 72.5 | 84.7 | 76.7 | 75.9 | 60.6 | – | 72.6 | 66.4 | 70.1 | 72.7 | 71.9 |
| MMBERT small | 71.4 | 77.0 | 85.8 | 79.9 | 79.0 | 67.9 | – | 75.6 | 72.6 | 75.3 | 73.2 | 75.8 |
| MMBERT base | **74.5** | **79.6** | **85.9** | **81.2** | **80.2** | **71.1** | – | **77.0** | **73.8** | **76.2** | **77.7** | **77.7** |
| **PAWS-X** | | | | | | | | | | | | |
| EuroBERT 210m | – | 87.8 | 94.9 | 89.6 | 90.0 | – | 77.6 | – | – | – | 80.2 | 86.7 |
| mmBERT small | – | 89.8 | 95.3 | **90.5** | 90.6 | – | 80.7 | – | – | – | 82.9 | 88.3 |
| mmBERT base | – | **90.7** | **95.8** | 90.2 | **92.0** | – | **81.8** | – | – | – | **83.6** | **89.0** |

Table 8: Comparing EuroBERT and MMBERT on **EuroBERT's in-distribution languages** (excluding others) on XNLI and PAWS-X. We see the MMBERT improves even on these languages. Dashes indicate a language not tested by the benchmark. Averages are for these languages only.

XLM-R). MMBERT outperforms the smaller variants: 73.6 XNLI vs Multilingual MiniLM's 71.2, even coming close to XLM-R's 74.6 despite having ~1/3 of the amount of non-embed parameters.

**Retrieval Results** We train all models on MS MARCO (English, §B) and evaluate on both English and multilingual benchmarks from MMTEB v2. We also evaluate on code using the CoIR benchmark.

We again see large gains in English MTEB v2 (Table 5) with even MMBERT small outperforming mGTE and XLM-R. MMBERT outperforms mGTE (the next closest) with an average of 53.9 vs 52.7, even performs similarly to ModernBERT's 53.8 average.

For multilingual MTEB v2 (Table 6), we find that both MMBERT's score about 1.5 points better on average than their similarly sized counterparts (i.e. 54.1 avg for MMBERT base vs XLM-R's 52.4).

On code retrieval tasks (CoIR, Table 7) we see that MMBERT performs significantly better than any other massively multilingual model (42.2 MMBERT base average vs XLM-R's 33.6), but underperforms compared to EuroBERT-210m (45.3 average). This is likely due to EuroBERT's use of the higher quality Stack v2 corpus, which we did not have access to.

**Overall, we see that MMBERT is an improved drop-in replacement for XLM-R**, and that even MMBERT small can come close to XLM-R's performance.

## 4.2 COMPARISON TO EUROBERT

As EuroBERT was only trained on 15 languages, it has a disadvantage on these massively multilingual evaluations and was excluded due to its low performance. Instead, we directly compare EuroBERT only on languages it was trained on. We show scores on these selected languages for XNLI and PAWS-X.[6] Table 8 shows that MMBERT base and small outperform EuroBERT-210m on these languages as well (i.e. 74.5 F1 on Arabic XNLI for MMBERT base vs 66.8 F1 for EuroBERT, etc.).

---

[6]Since the official EuroBERT model does not have a "ForQuestionAnswering" or "ForMultipleChoice" class available we select two tasks that work for classification.

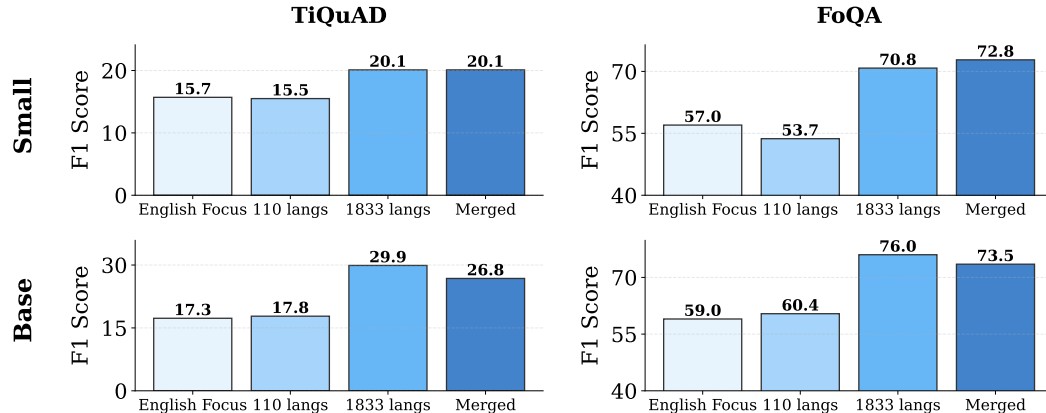

Figure 2: Performance of models using different decay phases on two languages (Tigrinya and Faroese) only added during the decay phase. We see that MMBERT with the 1833 language decay phase shows rapid performance improvements despite only having the models in the last 100B tokens of training. The final MMBERT models shows improvements by merging together checkpoints.

### 4.3 COMPARISON TO SIMILAR-SIZED SOTA DECODER MODELS

Although numerous previous works (Weller et al., 2025; Gisserot-Boukhlef et al., 2025) have shown that encoder-only models significantly outperform decoder models within the same size range (or even one order-of-magnitude bigger), we also test this by running the recent Gemma 3 270M (Team et al., 2025) model on the same classification tasks. Using the same hyperparameter sweep detailed previously, it scores 69.0 on XNLI and averages 82.9 on GLUE. Notably, this is much worse than even MMBERT small, again showing the benefits of encoder-only models for these tasks.[7]

### 4.4 ANNEALING LANGUAGE LEARNING

We test whether our decision to include more languages at the decay phase significantly improves model performance. We do this by selecting evaluation datasets that test the model on a language learned only during the decay phase. However, since these languages are low-resource there are not many high-quality evaluation datasets available. Thus, we test only two languages that have high quality evaluation data and are commonly[8] used: TiQuaD for Tigrinya (Gaim et al., 2023) and FoQA for Faroese (Simonsen et al., 2025). We evaluate in the same way as XQuAD (e.g. zero-shot from English SQuAD) and compare scores for the the different decay mixtures and the final merged model.

Figure 2 shows that there is a significant jump in performance for the variants that included the language: a 68% increase for base (12.1 absolute F1) on Tigrinya (110 to 1833 langs) and a 26% increase (15.4 absolute F1) for Faroese. On FoQA, which benchmarked larger LMs, it even outperforms Google's Gemini 2.5 Pro by 6 pts (69.8) and OpenAI's o3 by 8.3 points (67.7). Even MMBERT small outperforms these giant LMs.

Thus we find that annealing language learning significantly boosts low-resource language performance. We also see that the model merging allowed the model to retain most of its performance despite merging with English and higher-resource focused versions.

### 4.5 EFFICIENCY

MMBERT models are significantly faster than any previous multilingual encoder-only model. Figure 3 benchmarks this using the best settings available for each model using the transformers (Wolf et al.,

---

[7]We note that an embedding version of Gemma 3 270M, EmbeddingGemma, outperforms other models on MTEB, but crucially they fine-tune the model using a proprietary 320 billion token training set. They would likely see increased performance over GemmaEmbeddings if they simply fine-tuned MMBERT with the same data. However, as their data is not publicly available we cannot run this comparison.

[8]Are included in other benchmarks, i.e. ScandEval (Nielsen, 2023).

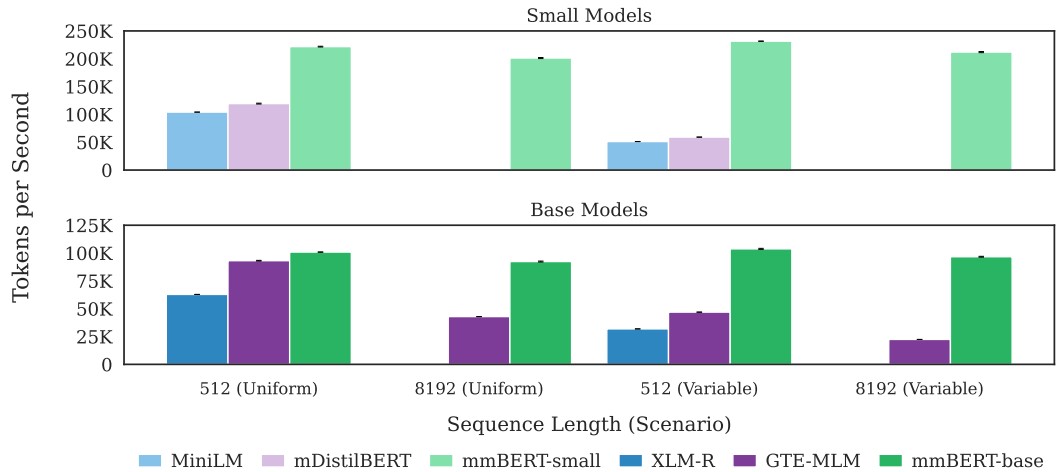

Figure 3: Throughput efficiency for various sequence lengths and variable input length (top row is small models, bottom row is base models). MMBERT is faster than previous models even when using 8x as many tokens, often 2-4x as fast. MMBERT models are more efficient because of the use of Flash Attention 2 and unpadding techniques it inherits from the ModernBERT configuration. Empty bars indicate that the model cannot use that sequence length (e.g. over 512 tokens). The error bars show standard error over five random seeds.

2020) implementation, e.g. by installing `xformers` or `flash-attention` when helpful. We find that MMBERT base is more than 2x faster on variable sequences and significantly faster on long context lengths ($\sim 4$x). For MMBERT small results are roughly 2x faster than MMBERT base and again roughly 2x faster than similarly-sized other multilingual models. We note that previous multilingual models such as Multilingual MiniLM and XLM-R cannot go past 512 tokens whereas MMBERT can perform at up to 8192 tokens – and can do so as fast as other models are at 512 tokens.

## 5 CONCLUSION

We introduce MMBERT, a modern multilingual encoder trained on 3T tokens and 1833 languages. We introduce several novel elements in training: an inverse masking schedule and a cascading annealed language learning schedule for multilingual data that progressively lower the language sampling temperature while also increasing the number of languages. MMBERT improves over the previous generation of multilingual encoders such as XLM-R while being a drop-in replacement. We further show that leaving out low-resource languages until the decay phase allows us to rapidly learn the languages on minute amounts of data, improving performance to levels past even SoTA large LMs like OpenAI's o3 and Google's Gemini 2.5 Pro. We open-source our models, data, and checkpoints to encourage future research and enable continued pre-training for new data and domains.

## 6 LIMITATIONS

Although MMBERT shows significant improvements on low-resource languages, there are still many languages that have very small amounts of data, or even none at all. This is especially notable for high-quality data, such as edu-style filtering (Lozhkov et al., 2024). We leave improvements in this area to future work, which would further improve scores on low-resource languages.

Due to the lack of higher quality data, we note that the data from FineWeb2 still contains noisy and biased data due to the large scale of the data (although much was removed thanks to their efforts). However, encoder-only models generally have reduced risks due to their lack of generative abilities, but still retain the ability to be biased in their predictions.

MMBERT models also show limited improvements over existing models on NER/POS tasks due to the lack of consistent `add_prefix_space` flag in the Gemma 2 tokenizer. Although we show some improvements, we would encourage future work to fix this for better improvements in these tasks.

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

## A ARCHITECTURE DETAILS

Common architecture details are in Table A. One other area of difference between the two models is that MMBERT small used a learning rate warmup of 4B tokens and a batch size warmup of 100B tokens. MMBERT base had a learning rate warmup of 3B tokens and a batch size warmup of 60B tokens.

## B HYPERPARAMETER AND COMPUTE DETAILS

**Compute**   We use a mix of H100 and L40s for training and inference, using the L40s mainly for the smaller model inference. We use 8xH100s for roughly 10 days to train the small version and 8xH100s for 40 days for the base version. For evaluation, each experiment takes roughly 1-2 hours to run for a given setting.

**NLU Sweep**   We sweep over seven LRs ({2e-5, 3e-5, 4e-5, 5e-5, 6e-5, 7e-5, 8e-5 } and four epoch options ({1, 2, 3, 5, 10}) using a batch size of 32 and a warmup ratio of 0.06 (following mGTE). We select the best result for each model on each task in an oracle fashion. Most models had the best score in the 2e-5 or 3e-5 range with varying amounts of epochs according to the task.

**Embedding Sweep**   We sweep four LRs, following Weller et al. (2025) {1e-4, 3e-4, 5e-4, 7e-4} and use the best one. We train with SentenceTransformers (Reimers & Gurevych, 2019) using 1.25M hard triplets for one epoch on MS MARCO data (Bajaj et al., 2016) from sentence-transformers/msmarco-co-condenser-margin-mse-sym-mnrl-mean-v1. We then evaluate using MTEB (Enevoldsen et al., 2025) and select the LR that performed the best. In all cases the best was using the 1e-4 LR (except for MiniLM which was 5e-4).

**FoQA and TiQuAD**   We use a partial sweep from the NLU section, using just the two LRs that performed best from the original sweep (2e-5, 3e-5) while also varying epochs.

| Parameter | Value |
|---|---|
| Learning Rate | 8e-4 |
| Weight Decay | 8e-5 |
| Training Batch Size | 4.7M tokens |
| Vocabulary Size | 256,000 |
| Max Sequence Length | 1024->8192 |
| Tokenizer | Gemma 2 |
| Attention Layer | RoPE |
| Attention Dropout | 0.0 |
| Attention Output Bias | false |
| Attention Output Dropout | 0.1 |
| Attention QKV Bias | false |
| Transformer Layer | prenorm |
| Embedding Dropout | 0.0 |
| Embedding Norm | true |
| Final Norm | true |
| Skip First PreNorm | true |
| MLP Dropout | 0.0 |
| MLP Input Bias | false |
| MLP Layer Type | GLU |
| MLP Output Bias | false |
| Normalization | LayerNorm |
| Norm Epsilon | 1e-12 |
| Norm Bias | false |
| Hidden Activation | GELU |
| Head Pred Activation | GELU |
| Activation Function | GELU |
| Padding | unpadded |
| Rotary Embedding Base | 10k -> 160k |
| Rotary Embedding Interleaved | false |
| Allow Embedding Resizing | true |
| Sliding Window | 128 |
| Global Attention Every N Layers | 3 |
| Unpad Embeddings | true |

Table 9: Common Configuration Parameters. Note that MMBERT small had to lower the LR/WD to half the initial value after 1.2T tokens, due it plateauing early.

## C  COMPARISON WITH DEBERTA

Both Ettin and ModernBERT found that comparable RTD-trained models show significantly worse performance on embedding tasks. We thus exclude them from our main analysis.

We validate this by running mDeBERTa on MTEB tasks. On the multilingual benchmark, we find it has an average of 42.5, more than 11 points worse than MMBERT's 54.1. mDeBERTa scores 48.6 on the English MTEB v2, again much worse than any comparable model (including our MMBERT small). Thus, RTD-trained models may do well at classification but they do so at the expense of embedding tasks which are a main use case of encoder-only models.

## D  LANGUAGE DISTRIBUTIONS

The data per language is found in Table 10. We use an inverse temperature sampling with values 0.7, 0.5, and 0.3 respectively. As only 90 rows would fit on the page, the full data is available on the Github in CSV form.

## E  LLM USAGE

LLMs were not used for paper writing, but were used for coding assistance and title brainstorming. All code was human verified.

| | Pretrain | | Mid-Training | | Decay-Eng | | Decay-Cont | | Decay-All | | |
|---|---|---|---|---|---|---|---|---|---|---|---|
| Lang | Value | % | Value | % | Value | % | Value | % | Value | % | Total |
| eng | 686.4 | 34.51% | 73.0 | 12.14% | 18.7 | 17.89% | 13.6 | 13.19% | 11.6 | 10.15% | 803.3 |
| code | 102.0 | 5.13% | 20.0 | 3.33% | 3.3 | 3.12% | 3.3 | 3.15% | 6.3 | 5.51% | 134.8 |
| rus | 115.4 | 5.80% | 30.2 | 5.02% | 2.8 | 2.72% | 3.1 | 2.98% | 3.0 | 2.61% | 154.5 |
| deu | 87.3 | 4.39% | 24.7 | 4.11% | 2.5 | 2.44% | 2.8 | 2.75% | 2.8 | 2.46% | 120.2 |
| spa | 89.8 | 4.51% | 25.2 | 4.19% | 2.5 | 2.38% | 2.7 | 2.62% | 2.7 | 2.37% | 122.8 |
| fra | 79.8 | 4.01% | 23.1 | 3.85% | 2.4 | 2.28% | 2.6 | 2.50% | 2.6 | 2.23% | 110.4 |
| cmn | 103.7 | 5.21% | 27.9 | 4.65% | 2.5 | 2.38% | 2.6 | 2.47% | 2.5 | 2.21% | 139.2 |
| ita | 57.3 | 2.88% | 18.3 | 3.05% | 2.0 | 1.96% | 2.2 | 2.12% | 2.1 | 1.88% | 82.0 |
| por | 46.9 | 2.36% | 16.5 | 2.75% | 1.9 | 1.83% | 2.0 | 1.98% | 2.0 | 1.76% | 69.4 |
| jpn | 58.6 | 2.95% | 18.6 | 3.10% | 2.0 | 1.87% | 2.0 | 1.96% | 2.0 | 1.75% | 83.2 |
| nld | 37.2 | 1.87% | 13.4 | 2.23% | 1.8 | 1.68% | 2.0 | 1.90% | 1.9 | 1.70% | 56.2 |
| pol | 36.5 | 1.84% | 13.2 | 2.20% | 1.7 | 1.65% | 1.9 | 1.86% | 1.9 | 1.68% | 55.3 |
| swe | 18.6 | 0.94% | 8.1 | 1.36% | 1.4 | 1.37% | 1.7 | 1.62% | 1.6 | 1.42% | 31.5 |
| ind | 32.5 | 1.63% | 12.2 | 2.03% | 1.5 | 1.47% | 1.6 | 1.56% | 1.6 | 1.41% | 49.4 |
| kor | 28.4 | 1.43% | 11.1 | 1.84% | 1.5 | 1.39% | 1.5 | 1.47% | 1.5 | 1.31% | 43.9 |
| arb | 21.2 | 1.07% | 9.0 | 1.50% | 1.4 | 1.30% | 1.5 | 1.43% | 1.7 | 1.48% | 34.7 |
| ces | 22.7 | 1.14% | 9.4 | 1.57% | 1.4 | 1.32% | 1.5 | 1.41% | 1.4 | 1.24% | 36.4 |
| tur | 24.9 | 1.25% | 10.1 | 1.68% | 1.4 | 1.32% | 1.4 | 1.40% | 1.4 | 1.25% | 39.3 |
| ukr | 17.5 | 0.88% | 7.8 | 1.30% | 1.3 | 1.22% | 1.4 | 1.39% | 1.4 | 1.25% | 29.4 |
| fas | 23.3 | 1.17% | 9.6 | 1.60% | 1.4 | 1.30% | 1.4 | 1.39% | 1.4 | 1.24% | 37.1 |
| vie | 22.2 | 1.12% | 9.3 | 1.54% | 1.3 | 1.27% | 1.4 | 1.37% | 1.4 | 1.23% | 35.6 |
| ron | 22.4 | 1.12% | 9.3 | 1.55% | 1.3 | 1.26% | 1.4 | 1.32% | 1.4 | 1.25% | 35.8 |
| nob | 20.9 | 1.05% | 8.9 | 1.48% | 1.3 | 1.22% | 1.3 | 1.30% | 1.3 | 1.17% | 33.8 |
| hun | 20.2 | 1.02% | 8.7 | 1.45% | 1.3 | 1.20% | 1.3 | 1.27% | 1.3 | 1.14% | 32.8 |
| fin | 14.9 | 0.75% | 7.0 | 1.16% | 1.2 | 1.12% | 1.3 | 1.24% | 1.3 | 1.10% | 25.5 |
| dan | 19.2 | 0.96% | 8.4 | 1.40% | 1.2 | 1.15% | 1.2 | 1.20% | 1.2 | 1.08% | 31.2 |
| tha | 18.2 | 0.92% | 8.1 | 1.34% | 1.2 | 1.14% | 1.2 | 1.19% | 1.2 | 1.06% | 29.9 |
| ell | 16.4 | 0.83% | 7.5 | 1.25% | 1.2 | 1.12% | 1.2 | 1.19% | 1.2 | 1.05% | 27.5 |
| bul | 12.9 | 0.65% | 6.3 | 1.05% | 1.0 | 0.99% | 1.1 | 1.04% | 1.1 | 0.93% | 22.4 |
| slk | 12.2 | 0.61% | 6.1 | 1.01% | 1.0 | 0.96% | 1.0 | 0.99% | 1.0 | 0.89% | 21.3 |
| hrv | 11.3 | 0.57% | 5.8 | 0.96% | 1.0 | 0.92% | 1.0 | 0.96% | 1.0 | 0.87% | 20.0 |
| srp | 0.0 | 0.00% | 3.8 | 0.63% | 1.0 | 0.92% | 1.0 | 0.93% | 0.9 | 0.82% | 6.7 |
| heb | 8.3 | 0.42% | 4.6 | 0.77% | 0.9 | 0.85% | 0.9 | 0.92% | 0.9 | 0.81% | 15.7 |
| hin | 10.0 | 0.50% | 5.3 | 0.88% | 0.9 | 0.87% | 0.9 | 0.90% | 1.2 | 1.04% | 18.3 |
| lit | 8.8 | 0.44% | 4.8 | 0.80% | 0.9 | 0.83% | 0.9 | 0.86% | 0.9 | 0.78% | 16.2 |
| ekk | 7.0 | 0.35% | 4.1 | 0.68% | 0.8 | 0.79% | 0.9 | 0.84% | 0.8 | 0.74% | 13.7 |
| slv | 7.9 | 0.40% | 4.5 | 0.74% | 0.8 | 0.80% | 0.9 | 0.83% | 0.8 | 0.74% | 14.9 |
| uzn | 0.0 | 0.00% | 2.4 | 0.40% | 0.8 | 0.80% | 0.8 | 0.81% | 0.7 | 0.63% | 4.8 |
| bos | 0.0 | 0.00% | 4.7 | 0.78% | 0.8 | 0.79% | 0.8 | 0.80% | 0.8 | 0.72% | 7.2 |
| ben | 6.5 | 0.33% | 3.9 | 0.64% | 0.8 | 0.74% | 0.8 | 0.79% | 0.9 | 0.78% | 12.8 |
| cat | 0.0 | 0.00% | 4.4 | 0.74% | 0.8 | 0.77% | 0.8 | 0.77% | 0.8 | 0.70% | 6.8 |
| zsm | 6.3 | 0.32% | 3.8 | 0.63% | 0.7 | 0.72% | 0.8 | 0.75% | 0.8 | 0.73% | 12.4 |
| lvs | 6.2 | 0.31% | 3.7 | 0.62% | 0.8 | 0.72% | 0.8 | 0.75% | 0.8 | 0.66% | 12.2 |
| azj | 4.8 | 0.24% | 3.1 | 0.52% | 0.7 | 0.66% | 0.7 | 0.69% | 0.7 | 0.65% | 10.0 |
| als | 0.0 | 0.00% | 2.9 | 0.49% | 0.7 | 0.65% | 0.7 | 0.65% | 0.6 | 0.54% | 4.9 |
| kaz | 3.0 | 0.15% | 2.2 | 0.37% | 0.6 | 0.56% | 0.6 | 0.61% | 0.6 | 0.54% | 7.0 |
| tam | 3.1 | 0.15% | 2.3 | 0.38% | 0.6 | 0.56% | 0.6 | 0.60% | 0.7 | 0.61% | 7.2 |
| urd | 3.6 | 0.18% | 2.6 | 0.43% | 0.6 | 0.58% | 0.6 | 0.60% | 0.9 | 0.74% | 8.3 |
| kat | 2.5 | 0.13% | 1.9 | 0.32% | 0.5 | 0.51% | 0.6 | 0.56% | 0.6 | 0.50% | 6.1 |
| isl | 0.0 | 0.00% | 2.1 | 0.35% | 0.6 | 0.53% | 0.6 | 0.53% | 0.5 | 0.44% | 3.7 |
| mkd | 2.6 | 0.13% | 2.0 | 0.33% | 0.5 | 0.50% | 0.5 | 0.53% | 0.5 | 0.47% | 6.2 |
| afr | 2.7 | 0.13% | 2.1 | 0.34% | 0.5 | 0.50% | 0.5 | 0.52% | 0.5 | 0.46% | 6.3 |
| tel | 2.1 | 0.10% | 1.7 | 0.29% | 0.5 | 0.48% | 0.5 | 0.52% | 0.6 | 0.49% | 5.4 |
| mya | 1.8 | 0.09% | 1.6 | 0.26% | 0.5 | 0.48% | 0.5 | 0.51% | 0.5 | 0.42% | 4.9 |
| ary | 0.0 | 0.00% | 2.1 | 0.35% | 0.5 | 0.51% | 0.5 | 0.51% | 0.5 | 0.45% | 3.7 |
| bel | 0.0 | 0.00% | 1.8 | 0.29% | 0.5 | 0.50% | 0.5 | 0.51% | 0.4 | 0.39% | 3.3 |
| mar | 2.6 | 0.13% | 2.0 | 0.34% | 0.5 | 0.48% | 0.5 | 0.50% | 0.5 | 0.48% | 6.2 |
| fil | 2.6 | 0.13% | 2.0 | 0.34% | 0.5 | 0.48% | 0.5 | 0.50% | 0.5 | 0.44% | 6.1 |
| glg | 2.2 | 0.11% | 1.8 | 0.30% | 0.5 | 0.47% | 0.5 | 0.49% | 0.5 | 0.44% | 5.5 |
| mal | 2.0 | 0.10% | 1.7 | 0.28% | 0.5 | 0.45% | 0.5 | 0.49% | 0.6 | 0.49% | 5.2 |
| npi | 2.5 | 0.13% | 2.0 | 0.33% | 0.5 | 0.47% | 0.5 | 0.49% | 0.5 | 0.44% | 6.0 |
| lat | 0.0 | 0.00% | 1.4 | 0.24% | 0.5 | 0.45% | 0.5 | 0.45% | 0.4 | 0.34% | 2.7 |
| bod | 0.0 | 0.00% | 0.8 | 0.13% | 0.5 | 0.45% | 0.5 | 0.45% | 0.2 | 0.21% | 1.9 |
| khk | 1.7 | 0.08% | 1.5 | 0.24% | 0.4 | 0.43% | 0.5 | 0.44% | 0.4 | 0.37% | 4.4 |
| pan | 0.0 | 0.00% | 1.2 | 0.20% | 0.4 | 0.43% | 0.4 | 0.44% | 0.4 | 0.33% | 2.5 |
| gmh | 0.0 | 0.00% | 1.6 | 0.26% | 0.4 | 0.43% | 0.4 | 0.43% | 0.4 | 0.37% | 2.9 |
| guj | 1.7 | 0.09% | 1.5 | 0.26% | 0.4 | 0.41% | 0.4 | 0.43% | 0.5 | 0.39% | 4.6 |
| anp | 0.0 | 0.00% | 1.6 | 0.27% | 0.4 | 0.42% | 0.4 | 0.42% | 0.4 | 0.38% | 2.9 |
| hye | 0.0 | 0.00% | 1.2 | 0.19% | 0.4 | 0.41% | 0.4 | 0.41% | 0.3 | 0.30% | 2.4 |
| rmy | 0.0 | 0.00% | 1.0 | 0.17% | 0.4 | 0.40% | 0.4 | 0.41% | 0.3 | 0.28% | 2.2 |
| eus | 0.0 | 0.00% | 1.4 | 0.23% | 0.4 | 0.40% | 0.4 | 0.40% | 0.4 | 0.34% | 2.6 |
| kan | 0.0 | 0.00% | 1.4 | 0.23% | 0.4 | 0.39% | 0.4 | 0.39% | 0.4 | 0.35% | 2.6 |
| cym | 0.0 | 0.00% | 1.1 | 0.18% | 0.4 | 0.39% | 0.4 | 0.39% | 0.3 | 0.30% | 2.2 |
| khm | 1.4 | 0.07% | 1.3 | 0.21% | 0.4 | 0.38% | 0.4 | 0.39% | 0.4 | 0.34% | 3.9 |
| swh | 1.3 | 0.06% | 1.2 | 0.20% | 0.4 | 0.37% | 0.4 | 0.38% | 0.4 | 0.33% | 3.7 |
| sin | 1.2 | 0.06% | 1.1 | 0.19% | 0.4 | 0.36% | 0.4 | 0.38% | 0.8 | 0.66% | 3.9 |
| ars | 0.0 | 0.00% | 1.2 | 0.20% | 0.4 | 0.37% | 0.4 | 0.37% | 0.4 | 0.32% | 2.3 |
| nno | 0.0 | 0.00% | 1.1 | 0.19% | 0.4 | 0.35% | 0.4 | 0.35% | 0.3 | 0.30% | 2.2 |
| bew | 0.0 | 0.00% | 1.1 | 0.19% | 0.4 | 0.35% | 0.4 | 0.35% | 0.3 | 0.30% | 2.2 |
| ory | 0.0 | 0.00% | 0.9 | 0.15% | 0.4 | 0.34% | 0.4 | 0.35% | 0.3 | 0.26% | 1.9 |
| kir | 0.0 | 0.00% | 1.0 | 0.17% | 0.4 | 0.34% | 0.4 | 0.34% | 0.3 | 0.28% | 2.1 |
| arz | 0.0 | 0.00% | 1.1 | 0.18% | 0.3 | 0.33% | 0.3 | 0.33% | 0.3 | 0.30% | 2.1 |
| gle | 1.0 | 0.05% | 1.0 | 0.17% | 0.3 | 0.32% | 0.3 | 0.33% | 0.3 | 0.30% | 3.0 |
| tgk | 0.0 | 0.00% | 1.0 | 0.17% | 0.3 | 0.32% | 0.3 | 0.33% | 0.3 | 0.28% | 2.0 |
| som | 0.0 | 0.00% | 1.0 | 0.17% | 0.3 | 0.32% | 0.3 | 0.32% | 0.3 | 0.28% | 2.0 |
| amh | 0.0 | 0.00% | 0.7 | 0.11% | 0.3 | 0.32% | 0.3 | 0.32% | 0.2 | 0.21% | 1.6 |
| pbt | 0.9 | 0.04% | 0.9 | 0.16% | 0.3 | 0.31% | 0.3 | 0.32% | 0.3 | 0.29% | 2.8 |
| gsw | 0.0 | 0.00% | 0.9 | 0.15% | 0.3 | 0.31% | 0.3 | 0.32% | 0.3 | 0.25% | 1.8 |
| tat | 0.0 | 0.00% | 0.9 | 0.15% | 0.3 | 0.31% | 0.3 | 0.31% | 0.3 | 0.29% | 1.9 |
| hif | 0.0 | 0.00% | 0.8 | 0.14% | 0.3 | 0.30% | 0.3 | 0.31% | 0.3 | 0.26% | 1.8 |
| **Total** | 1989.0 | 100.00% | 600.8 | 100.00% | 104.4 | 100.00% | 103.3 | 100.00% | 114.2 | 100.00% | 2911.8 |

Table 10: Language data (in billions, rounded to nearest 100M) with stage percentages for the first 90 language included. More languages would not fit on the page, see the Github for the full CSV details.

