# OpenReview forum: "mmBERT: A Modern Multilingual Encoder with Annealed Language Learning"
_ICLR.cc/2026/Conference — Submitted to ICLR 2026_

### Official Review · Reviewer_QX8Z · 2025-10-30

**Soundness:** 2
**Presentation:** 3
**Contribution:** 2
**Rating:** 2
**Confidence:** 1

**Summary:**

This paper introduces a new multilingual encoder-only language model called mmBERT. The authors train it with a novel annealing method that increases the uniformity of language sampling throughout training -- this should help the model to support over 1800 languages without overfitting on the low-resource ones.

**Strengths:**

The paper comes with the release of two language models that might be very useful for NLP practitioners around the world. This is a great strength of this paper, especially considering the cost of training those models on 3T tokens. The performance of these models seems very good and they should be very efficient to run thanks to the ModernBERT architecture they are based on.

**Weaknesses:**

My main issue is that the paper has no real scientific contributions. The paper mentions 3 pretraining contributions: *"(1) an inverse mask schedule learning rate (high ⇒low), (2) an annealing language schedule (more biased ⇒more uniform), and (3) increasing the number of languages at each training phase (60→110→1833), allowing for maximal impact of the smaller amount of data."* -- but their relevance is greatly overstated:
1) The first contribution is a large problem: This paper plagiarizes "Dynamic Masking Rate Schedules for MLM Pretraining" [1]. That paper introduces exactly the same method as advertised here; as with exactly the same hyperparameters, they decay the masking rate from 30% to 15%. Yet, this paper is never cited here and the method is even presented as a novel contribution of this paper.
2) The second contribution sounds interesting and makes intuitive sense, however it is never empirically evaluated in this paper. It might be possible that the observed performance improvements come from the high-quality data or improved transformer architecture, not from the "annealing language schedule". The Result's subsection 4.4 "Annealing language learning" seems evaluating this method, but in fact it only checks whether adding new languages at the of training improves the performance on these languages -- which it obviously does when compared to a baseline that has never seen those languages. But this is not a proper evaluation of the contribution 2); the baseline used in previous work is to have constant sampling throughout the whole training, the proposed method should be compared to that.
3) The third contribution is actually the same as the second one, it just means that the language schedule start at 0 for some languages.

Other than that, the contribution is the release of mmBERT, which I noted as a clear strength, but that alone does not bring any new scientific finding.

**Some other mistakes and issues:**
- The paper completely ignores the existence of mDeBERTa (coincidentally an ICLR paper) [2] and claims that "XLM-R is still SOTA" without any support. It is not true in practice as well as according to [2]. The bigger issue though is that mDeBERTa is missing from the evaluation, which might exaggerate the performance of the model proposed in this paper.
- There has been important work in multilingual language modeling that is completely ignored by the authors, in particular the UniMax sampling of languages used for training the umT5 models [3], as well as the modular multilingual mmT5 [4].
- *"encoder-only language models cannot generate text"* (line 31) -- that's clearly not true, as demonstrated by [5, 6] and as demonstrated by the countless recently popular masked diffusion language models.
- *"However, data quality has significantly increased since 2019, allowing us to achieve higher scores with only half of the tokens (Penedo et al., 2024)."* (line 96) -- this claim is true for causal-only models, not for encoder-only models (used in this paper) that have a completely different training behavior.
- Saying *"we achieve similar classification performance to models like OpenAI’s o3 and Google’s Gemini 2.5 Pro"* is a clear overstatement over-selling this work. It is a well-known fact that finetuning BERT-like models on in-domain data can outperform LLMs evaluated zero-shot. It is not a special property of this particular model and it is not a fair comparison.

____

**References:**
- [1] Dynamic Masking Rate Schedules for MLM Pretraining -- https://aclanthology.org/2024.eacl-short.42/
- [2] DeBERTaV3: Improving DeBERTa using ELECTRA-Style Pre-Training with Gradient-Disentangled Embedding Sharing
 -- https://openreview.net/forum?id=sE7-XhLxHA
- [3] UniMax: Fairer and More Effective Language Sampling for Large-Scale Multilingual Pretraining -- https://openreview.net/forum?id=kXwdL1cWOAi
- [4] mmT5: Modular Multilingual Pre-Training Solves Source Language Hallucinations -- https://aclanthology.org/2023.findings-emnlp.132/
- [5] BERT has a Mouth, and It Must Speak: BERT as a Markov Random Field Language Model -- https://aclanthology.org/W19-2304/
- [6] BERTs are Generative In-Context Learners: https://proceedings.neurips.cc/paper_files/paper/2024/hash/04ea184dfb5f1babb78c093e850a83f9-Abstract-Conference.html

**Questions:**

- Why did you choose to use the tokenizer from Gemma 2? Wouldn't it be more appropriate to train a new tokenizer on the specific distribution of languages you're using? This way, the tokenization (and thus the whole model) might be very inefficient for language underrepresented in the Gemma 2 tokenizer (it's unclear how many languages are supported by Gemma 2, but the subsequent Gemma 3 only supports 140+ languages, an order of magnitude less than mmBERT). This seems to defeat the main purpose of your multilingual language model.
- You have included a substantial amount of programming code in the data mixture, is it treated as "English" language, as "code" language, or in some other way?

---

> ### Author Response · Authors · 2025-11-18
>
> Thank you for recognizing that our work could be “very useful for NLP practitioners around the world”!
>
> Overall we would like to clarify some misunderstandings regarding our paper and address your concerns.
>
> > The paper completely ignores the existence of mDeBERTa (coincidentally an ICLR paper) [2] and claims that "XLM-R is still SOTA" without any support. It is not true in practice as well as according to [2]. The bigger issue though is that mDeBERTa is missing from the evaluation, which might exaggerate the performance of the model proposed in this paper.
>
> **We do directly compare with mDeBERTa**! Please see footnote 5 on page 6 (which points to Appendix C). RTD style models are terrible at embedding-based tasks (mDeBERTa is 11 absolute points worse than mmBERT!) which is a core component of encoders, which is why we left it out of the main paper. We know of no work using RTD-trained models for embeddings, which is why we do not include it as a SoTA general-purpose encoder.
>
> > The first contribution is a large problem: This paper plagiarizes "Dynamic Masking Rate Schedules for MLM Pretraining" [1]
>
> This is relevant related work and we will add a citation and discussion - **we certainly did not intend any plagiarism!** While it is relevant related work, there are many key differences. That work uses linearly decreasing mask rates through training, while we use a step function. We decay much more than they do, going to a 5% masking rate instead of 15%. We also change the underlying data distribution - each step stage is also a data change. That work uses C4 as a primary corpus, which is both much smaller and less diverse than our dataset (both in terms of language and content). Furthermore, we use both a longer base sequence length and change the sequence length over the course of training (1024 → 8192, vs static 128). These and other training differences (such as our WSD LR schedule vs decay), that we chose based on recent encoder training best practices, **make the setup explored in their work fairly different.** However we find it very promising that their result is directionally consistent with our setup and **will certainly include it as a citation!** Thank you for drawing our attention to the missing reference.
>
> > 2); the baseline used in previous work is to have constant sampling throughout the whole training, the proposed method should be compared to that
>
> We agree this would be nice, however, scaling to 3T tokens is quite compute intensive. We, like almost any other research group, do not have the compute to be able to do this many times for each new design decision.
>
> However, as you mention, our approach “sounds interesting and makes intuitive sense”, and as seen by our results, works very well. These strong results help validate our decisions. We also believe that our descriptions of how to achieve this model (plus the release of a new open-source SoTA model including the data!) is still a significant contribution to the community. We hope more people will investigate this topic.
>
> > "encoder-only language models cannot generate text" (line 31) -- that's clearly not true, as demonstrated by [5, 6] and as demonstrated by the countless recently popular masked diffusion language models.
>
> This is true that they can be forced into generation, but we think most people would agree that they have not achieved comparable generation quality to decoders. We will update this to change it to “encoder-only language models are not primarily trained for text generation (and generally do worse)”. We merely aimed to communicate to the reader why encoder-only LMs have become less popular due to lack of simple usage for text generation.
>
> (continued below)

---

> > ### Author Response · Authors · 2025-11-18
> >
> > (continued from above)
> >
> > > "However, data quality has significantly increased since 2019, allowing us to achieve higher scores with only half of the tokens (Penedo et al., 2024)." (line 96) -- this claim is true for causal-only models, not for encoder-only models (used in this paper) that have a completely different training behavior.
> >
> > Which part of our claim do you disagree with? We trained for half the tokens as XLM-R (which used 6T tokens) and did better.
> > Encoder-only models can be trained on the same data as decoder-only models (e.g. we did use existing data that has been used to train SmolLM and Olmo which are decoders). Furthermore, ModernBERT [1] showed that using decoder-only tricks work for encoders, and other work [2] showed that encoders do better on classification/retrieval than decoders with the same data.
> > Although they may seem separate, the only difference between them is the training objective (MLM vs NTP) and attention (bidirectional vs causal). For both, data quality is critical.
> >
> > [1] https://arxiv.org/abs/2412.13663
> >
> > [2] https://arxiv.org/abs/2507.11412
> >
> > > Saying "we achieve similar classification performance to models like OpenAI’s o3 and Google’s Gemini 2.5 Pro" is a clear overstatement over-selling this work. It is a well-known fact that finetuning BERT-like models on in-domain data can outperform LLMs evaluated zero-shot. It is not a special property of this particular model and it is not a fair comparison.
> >
> > We agree that this property is not unique to our model. **Nonetheless, our claim is still true** that our very small models can beat these larger models zero-shot – which some people are still unaware of. However, we will add language to clarify this point.
> >
> > > There has been important work in multilingual language modeling that is completely ignored by the authors
> >
> > Thank you for these references, we will cite them.
> >
> > > You have included a substantial amount of programming code in the data mixture, is it treated as "English" language, as "code" language, or in some other way?
> >
> > We label it as general “code” in our tables and figures (e.g. Figure 1, Table 10). For the LM, it sees them all the same as there are no language tags.
> >
> > > Why did you choose to use the tokenizer from Gemma 2? Wouldn't it be more appropriate to train a new tokenizer
> >
> > Training a new tokenizer involves a large amount of risk. We chose to de-risk by using an existing multilingual tokenizer that already covers many scripts. Although you are right that Gemma 2 does not officially support all languages we cover, there is a large amount of unofficial coverage due to shared scripts and similar languages. This makes it so the model can generalize well. However, we agree that new tokenizers would likely be a fruitful area of future work. See our response to reviewer dBWC for additional notes about tokenizer choice!
> >
> > > Other than that, the contribution is the release of mmBERT, which I noted as a clear strength, but that alone does not bring any new scientific finding.
> >
> > As mentioned previously, **we have many novel methods that contributed to a much stronger model than any previously created**. Although these could not be ablated due to the scale, our paper shows how to achieve such scores and provides the ability for others to improve on it. No lab (including industry groups) can perform ablations at their largest model scale.
> > We thus follow much previous work (e.g. ModernBERT, published at ACL, among many others) in documenting our novel steps and showing how to create such a model. In many ways go further than previous work, by publicly releasing our training data whereas most do not.

---

### Official Review · Reviewer_Vo7i · 2025-10-31

**Soundness:** 3
**Presentation:** 3
**Contribution:** 3
**Rating:** 8
**Confidence:** 3

**Summary:**

The paper introduces mmBERT, a modern multilingual encoder-only language model trained on 3T tokens across 1800+ languages. The core contributions are: In terms of specific training mechanisms the authors use: inverse mask ratio schedule (30% → 15% → 5%) and a cascading annealed language learning to progressively add languages during learning. Finally, model merging across different decay phase variants is used to improve final model. In experiments, mmBERT demonstrates competitive performance on benchmarks.

**Strengths:**

The annealed language learning is well-motivated and seems to be useful. In general, the large amount of language covered is a nice strength of the paper.
mmBERT shows trong empirical results across diverse benchmarks
Overall, this is a meaningful contribution to an underserved area (multilingual encoders)

**Weaknesses:**

The authors acknowledge they couldn't ablate inverse masking due to compute, but this is an interesting contribution that could be tested more directly.

Only 2 languages are tested for demonstrating the benefit of the annealed language learning contribution. In general, with many languages covered it is difficult to assess whether the encoder really does a decent job on all languages and the evaluation over the long tail of languages is very superficial.

The tables of results do not report variability making it impossible to assess when and whether improvements are statistically significant. This would be important for rigorous science to at least report some notion of the variability (std, confidence intervals)

**Questions:**

How were the hyperparameters choosen, especially for masking ratio and ALL?

---

> ### Author Response · Authors · 2025-11-18
>
> Thank you for recognizing that our work is “a meaningful contribution to an underserved area” and is “well-motivated” and “useful”!
>
> > The tables of results do not report variability making it impossible to assess when and whether improvements are statistically significant. This would be important for rigorous science to at least report some notion of the variability (std, confidence intervals)
>
> We followed previous work in reporting the maximum score over a large hyperparameter sweep for evaluation tasks, and unfortunately we did not retain all intermediate results. Given the size of the sweep and best practices in using encoders, we believe each result still represents achievable task performance. However, given such a large sweep, each model is given chances for random variability to work in their favor.
>
> > Only 2 languages are tested for demonstrating the benefit of the annealed language learning contribution. In general, with many languages covered it is difficult to assess whether the encoder really does a decent job on all languages and the evaluation over the long tail of languages is very superficial.
>
> We agree with this sentiment: evaluations are super important! However, there don’t exist many benchmarks at all for these tail languages which is why we selected just two that had high quality benchmarks used in the community. We hope to see more work for creating new benchmarks for low-resource languages, but it is out of scope for our work.

---

> > ### Comment · Reviewer_Vo7i · 2025-11-24
> >
> > Thank you to the authors for their answer. I will keep my positive assessment.
> >
> > I just mention that my comment about variability and confidence intervals does not refer to the variance coming from random seed in retraining (which is clearly difficult to estimate for large model) or hyper-parameters choices in finetuning (which would be nice to try and estimate) but about the variability of performance metrics on test data, which is easily estimated via, e.g., bootstrap resampling of the test set, or even just reporting the std of the mean to give an idea of the variance.

---

### Official Review · Reviewer_SM4k · 2025-11-01

**Soundness:** 2
**Presentation:** 2
**Contribution:** 3
**Rating:** 4
**Confidence:** 4

**Summary:**

This paper presents mmBERT, an encoder-only language model trained on 3T tokens across over 1800 languages, introducing training strategies such as an inverse mask ratio schedule and an annealed language learning schedule. The authors progressively add more languages during training, focusing on low-resource ones in the final phase. mmBERT shows substantial improvements over previous models (e.g., XLM-R, mBERT), matching or beating strong models in classification and retrieval tasks, including low-resource languages. The model and data are claimed to be open-source, supporting reproducibility and further research.

**Strengths:**

- trained to support large set of languages (1800+)
- Outperforms previous encoder models (XLM-R, EuroBERT)
- Method and training strategies are clearly described
- The model and data are claimed to be fully open-sourced
- strong multilingual encoder model released as backbone replacing xlm-r etc.

**Weaknesses:**

1. The effectiveness of the inverse masking schedule and cascading annealed language learning schedule are not well demonstrated through ablation studies. If these methods are claimed as novel contributions, it's important to thoroughly prove their effectiveness.
2. It is unclear how mmBERT compares to using a small decoder-only LLM like Qwen3-0.6B as a backbone for text embedding and classification. (If I recall correctly, decoder-only LLMs can enable bidirectional attention during downstream fine-tuning.) It is important to know whether the encoder architecture truly outperforms a decoder-only architecture. Since the models are trained from scratch, it would be ideal to train both architectures and compare them.
3. Although evaluating all 1800+ languages would be impractical, it would be better to present the model's performance on a selected set of languages covering different linguistic characteristics.

**Questions:**

- How does the inverse masking schedule compare to a fixed masking ratio under the same training strategies?
- How does the annealed language learning schedule compare to training the entire corpus together with low-resource languages upsampled?
- How does mmBERT compare to using a modern decoder-only architecture as the backbone?

---

> ### Author Response · Authors · 2025-11-18
>
> Thank you for recognizing our model “outperforms previous encoder models (XLM-R, EuroBERT)” with “clearly described” methods and is “fully open-sourced”!
>
> > It is unclear how mmBERT compares to using a small decoder-only LLM like Qwen3-0.6B as a backbone for text embedding and classification .. it is important to know whether the encoder architecture truly outperforms a decoder-only architecture
>
> **We agree and have already done this experiment to a SoTA decoder**! See Section 4.3 titled “Comparison to Similar Sized Decoder-Only Models” where we compare to the recent Gemma 3 270M. We significantly outperform them. This finding that encoders are better than decoders for these tasks has also been shown in the literature, see references in Section 4.3.
>
> > The effectiveness of the inverse masking schedule and cascading annealed language learning schedule are not well demonstrated through ablation studies. If these methods are claimed as novel contributions, it's important to thoroughly prove their effectiveness.
>
> We agree that this is less than ideal. However, at this scale (3T tokens), is it simply not possible to run ablations. However, our results show significantly improved performance, which does vouch for the quality of our methods.
>
> > Although evaluating all 1800+ languages would be impractical, it would be better to present the model's performance on a selected set of languages covering different linguistic characteristics
>
> **We agree and have also done so**! Each of our evals cover a wide range of languages, e.g. XNLI covers French, Spanish, German, Greek, Bulgarian, Russian, Turkish, Arabic, Vietnamese, Thai, Chinese, Hindi, Swahili and Urdu, while TyDiQA (**QA**
> in **Ty**pologically **Di**verse Languages) covers English, Bengali, Korean, Telugu, Swahili, Russian, Finnish, Indonesian, and Arabic.  If it would be helpful, we can include tables in the appendix showing the breakdown for these languages.
>
> We hope this may have cleared up any misunderstandings and please let us know if you have any other questions!

---

> > ### Comment · Reviewer_SM4k · 2025-11-27
> >
> > Regarding "Comparison to Similar Sized Decoder-Only Models," I apologize for overlooking that section in my initial reading.
> > As for the suggestion about "presenting the model's performance on a selected set of languages covering different linguistic characteristics," TyDiQA was already widely used during the XLM-RoBERTa era. I was hoping to see insights beyond the previously discussed languages—for instance, the long-tail languages among the 1,800 covered. However, I recognize this may be outside the scope of this work, given that the primary focus is on training methodology. This is merely a question rather than a concern.
> >
> > My major concern remains the following:
> >
> > **The effectiveness of the inverse masking schedule and cascading annealed language learning schedule is not well demonstrated.**
> >
> > I genuinely appreciate the release of the artifacts, and I find the training recipe for the released model to be clear. I also understand that computational cost can limit the feasibility of conducting exhaustive experiments.
> > However, as a scientific paper rather than a technical report, I believe the claimed contributions should be rigorously validated. If a researcher were to train an encoder model from scratch in the future, they would likely want to learn from clear conclusions about how effective the inverse masking schedule and cascading annealed language learning schedule truly are. In its current form, the paper does not provide sufficient evidence for researchers to confidently adopt these techniques. I still hope that more solid evidence can be provided to support the proposed contributions.

---

### Official Review · Reviewer_dBWC · 2025-11-10

**Soundness:** 3
**Presentation:** 3
**Contribution:** 3
**Rating:** 6
**Confidence:** 4

**Summary:**

This paper proposes MMBERT which introduces a few techniques from modern decoder-only LLMs into encoder-only BERT style models at large scale for 1800 languages.

**Strengths:**

• Large scale experiments: 3T tokens with 1800 languages
	• A 3-phrase recipe for training MMBERT: (1) inverse mask schedule; (2) annealing language schedule; (3) increasing number of languages in gradually in 3 phrases.

**Weaknesses:**

• Mostly known tricks; locating desired combination of them.

**Questions:**

Any ablation study on the vocab size? As scaling to 1800 languages, how severe is the presence of "unk" over the 3T token dataset especially on the languages of on glyph scripts?

---

> ### Author Response · Authors · 2025-11-18
>
> Thank you for noticing that our work shows “large scale experiments”!
>
> > • Mostly known tricks; locating desired combination of them.
>
> We believe **we did introduce novel tricks that are not known**: e.g. language annealing (sampling different amounts per language) and adding new languages. If you think these are known tricks, would you mind sharing where you saw them?
> We strongly agree that figuring out the right combination of things to use is a difficult part of creating such a model and we appreciate your recognition of that.
>
> > Any ablation study on the vocab size?
>
> We chose the vocab size to match Gemma 2 as it had the highest fertility on a small sample among existing tokenizers that were available at the time of this work. Sadly due to cost, we could not train the model twice with different tokenizers to compare the differences.
>
> > As scaling to 1800 languages, how severe is the presence of "unk" over the 3T token dataset especially on the languages of on glyph scripts?
>
> Good question! To illustrate this we checked the prevalence of  the unk token in a 100B sample of the initial pre-training (where all languages are covered by the Gemma 2 tokenizer) and the final decay stage with all 1800+ languages.
>
> | Model | Total Unknown Tokens | Samples with Unknown Tokens | Total Samples | Total Tokens | Unknown Token Rate | % Samples with Unknowns |
> |-------|---------------------|------------------------------|----------------|--------------|------------------|-------------------------|
> | Pre-training | 1,517 | 789 | 161,375,300 | 100,003,084,573 | $1.5\times10^{-8}$ | 0.0005% |
> | Decay Stage | 298 | 164 | 173,486,596 | 116,565,321,541 | $2.6\times10^{-9}$ | 0.0001% |
>
> You can see that there are more unk tokens in the pre-training stage (likely due to comparatively lower quality data) than there are in the decay stage even though those languages aren’t officially supported (they are supported unofficially through the use of similar scripts). But in both settings, there are a very small number of sequences with unknown tokens (0.0005% for pre-training and 1/5 of that for the decay) - sufficient coverage given the broad language support. *We can include these statistics in a revised version of the paper, thanks!*
>
> Thus, although we believe further tokenizer improvements would be very helpful for increasing performance, we did not focus on them (or claim them) in this work.

---

### Meta-Review · Area_Chair_XuPZ · 2026-01-06

**Summary:**

This paper presents mmBERT, a multilingual encoder-only language model trained on 3T tokens across 1800+ languages, introducing an inverse mask ratio schedule and annealed language learning approach. The reviewers expressed divergent views, with primary concerns centered on insufficient ablation studies for the proposed training strategies and questions about novelty, while acknowledging the significant practical value of releasing a strong multilingual encoder with open data [Reviewers SM4k, QX8Z, Vo7i, dBWC].

**Reviewer Concerns:**

**Addressed by Rebuttal:**
- The claim that mDeBERTa comparison was missing was addressed—authors clarified it exists in footnote 5 and Appendix C, showing RTD models perform poorly on embedding tasks (11 points worse) [Reviewer QX8Z]
- The decoder-only comparison concern was addressed—Section 4.3 already compares to Gemma 3 270M with significant improvements [Reviewer SM4k]
- The missing citation for dynamic masking schedules was acknowledged and will be added, though authors note methodological differences (step function vs. linear decay, 5% vs. 15% final rate, data distribution changes). **Though these may considered as very incremental contributions** [Reviewer QX8Z]

**Outstanding Concerns:**
- The effectiveness of inverse masking and annealed language learning schedules lacks rigorous ablation validation at scale [Reviewers SM4k, Vo7i]
- Evaluation coverage for long-tail languages remains superficial given 1800+ languages claimed [Reviewer Vo7i]
- No variability metrics (std, confidence intervals) reported for results [Reviewer Vo7i]
- **Limited or very incremental contributions** compare to the related work "Dynamic Masking Rate Schedules for MLM Pretraining"

**Reviewer Scores:**

Reviewer dBWC: 6. The reviewer may maintain their score, as their primary concern about "mostly known tricks" was partially addressed by clarifying the novel combination and language annealing approach, though ablation limitations persist.

Reviewer SM4k: 4. The reviewer may remain at marginally below threshold, as they explicitly stated the lack of ablation evidence for claimed contributions remains their "major concern" even after rebuttal engagement.

Reviewer Vo7i: 8. The reviewer explicitly stated they would "keep my positive assessment," acknowledging the meaningful contribution to an underserved area despite noting variability reporting limitations.

Reviewer QX8Z: 2. Given their lowest confidence score (1) and factual errors in their review (claiming missing mDeBERTa comparison that exists), the score may have increased slightly, though concerns about novelty claims would likely persist.

---

### Decision · Program_Chairs · 2026-01-26

Reject